# Exploring Occupational Health and Safety Standards Compliance in the South African Mining Industry, Limpopo Province, Using Principal Component Analysis

**DOI:** 10.3390/ijerph191610241

**Published:** 2022-08-18

**Authors:** Livhuwani Muthelo, Tebogo Maria Mothiba, Nancy Rambelani Malema, Masenyani Oupa Mbombi, Peter Modupi Mphekgwana

**Affiliations:** 1Department of Nursing Science, University of Limpopo, Mankweng 0727, South Africa; 2Faculty of Health Science, University of Limpopo, Mankweng 0727, South Africa; 3Department of Research Administration and Development, University of Limpopo, Mankweng 0727, South Africa

**Keywords:** health and safety, legislation, standards, miners

## Abstract

The health and safety of the miners in the South African mining industry are guided by the regulations and standards applied to promote a healthy work environment. The miners must comply with these regulations/standards to protect themselves from potential occupational health and safety risks, accidents, and fatalities. The status of compliance to safety regulations and standards in the mining industry of Limpopo Province has received little attention from scholars. This study explores the practices related to occupational health and safety standards compliance in the mining industry. A total of 277 miners were randomly selected from 1300 respondents in the mining industry. Data were collected using a 31-item survey questionnaire, administered to miners to explore occupational health and safety standards compliance from December 2019 to May 2020. Principal Component Analysis (PCA) extracted key attributes of occupational health and safety standards compliance in the mining industry and uncovered relationships between different dimensions. The study revealed that seven factors could measure occupational health and safety standards practices. It was observed that Factor 1 (occupational health practice related to regulations) is correlated with Factor 2 (measures to reduce risk of injuries/accidents). Additionally, Factor 2 (measures to minimise the risk of injuries/accidents) is correlated with Factor 4 (impact of the environment and production). There is a correlation between non-compliance with the safety regulations and the occurrence of injuries and accidents.

## 1. Introduction

Occupational health and safety standards play an essential role in guiding all miners on health and safety-related issues, focusing more on prevention. There is evidence of models, legislation, standards, and theories to mitigate the high incidences of accidents and fatalities in the mining industries [1,2]. Most health and safety authorities in mining companies worldwide agree that the major causes of mine accidents and fatalities are unsafe conditions and poor management, and especially non-compliance with health and safety standards [3,4]. The implementation of occupational legislation and standards is perceived as a significant challenge due to the miners’ lack of knowledge of occupational legislation, and adherence is, thus, impaired [5].

In Africa, there are comprehensive laws regarding health and safety standards and systems to ensure compliance; however, adherence to them is generally weak and under-resourced in many organisations [5,6]. The major challenges that contribute to poor safety compliance were identified as a poor coordination of activities in many organisations, a lack of specific health and safety regulations, and non-compliance with relevant H&S legislation [7,8]. The South African mining industry is legislated by regulations/standards and guidelines that guide both the employer and employee to maintain a safe and healthy environment. The legislation approved in the South African mining industry assists in the transformation and improvement of safety standards in the mining sector [9]. The miners’ health and safety standards are fundamental justiciable socio-economic rights [9]. Though there are legislations, policies, and standards in place, many reported health and safety incidents are related to poor compliance, including occupational injuries and illnesses, worldwide [6,10,11]. To close the gap, in 2016, the chamber of mines signed a declaration of actions pledge as an undertaking to improve the mining industry’s occupational health and safety to zero harm by 2024, and to generate a culture change in the industry that will transform the behaviour of people at all levels [12]. Still, the South African mining industry has not gained a respected reputation for health and safety due to repetitive mishaps and fatalities. The South African mining industry has gained a reputation for poor health and safety statistics, with non-compliance by miners and the executives being the major contributors to fatalities [13,14]. 

Promoting occupational health and safety in the workplace has several benefits for society; the mining organisations; the miners and their families; and South Africa’s (SA) socio-economic growth. Previous studies have reported that poor enforcement and implementation of health and safety measures negatively affect social and economic growth [2,11]. Several scholars identified a gap in the literature on the management of compliance with health and safety legislation [1,15]. The above background sparked the interest of the present authors to explore occupational health and safety standards compliance in the SA Mining Industry, Limpopo Province. 

## 2. Methodology

### 2.1. Study Context

A quantitative, descriptive, cross-sectional design was used to obtain data from the miners regarding compliance with health and safety standards [16]. The study was conducted at the selected mine, an open-cast mine based in the Mopani district, Limpopo Province. The mine produces phosphoric acid and phosphate-based granular fertilisers for local and international markets. Moreover, the mine has an on-site occupational health clinic that caters to the well-being of the miners.

### 2.2. Population and Data Collection

A total of 297 miners were conveniently sampled from 1300 miners. The sample constituted of miners, artisans, mining engineers, operators, electricians, shift bosses, team leaders, winch operators, health and safety personnel, mining managers, engineering managers, and mine surveyors in the mining industry. Exactly 277 miners returned the signed consent form, giving a 93% response rate from the initial sampled group of miners. The respondents who consented to participate were asked to complete the self-administered questionnaire written in English and three indigenous languages: Sepedi, Xitsonga, and Tshivenda. Data were collected for six months from December 2019 to May 2020, using a self-administered questionnaire with closed-ended questions (Refer to Appendix A). The questionnaire was structured into two sections: Section A—Demographic information such as gender, age, marital status, years of experience, and educational qualifications; and Section B—occupational health and safety standards compliance regulations. The questionnaire had clear instructions for the respondents to choose the number that best described their opinion of each item on the scale.

### 2.3. Data Analysis

Data were entered into Microsoft Excel and then exported to Statistical Package for the Social Sciences (SPSS) version 26 (IBM, Armonk, NY, USA). Demographic characteristics of the participants were presented as frequencies and percentages. Principal Component Analysis (PCA) was used to extract key attributes from 29 items related to occupational health and safety standards practices in the mining industry, and to uncover relationships between the different dimensions. A Kaiser–Meyer–Olkin measure of >0.5 means that the sample size was appropriate to run an Exploratory Factor Analysis (EFA) [17]. Only items with factor loadings of more than 0.30 were retained [16]. The performance of the PCA commenced with the Kaiser–Meyer–Olkin test for compliance with occupational health and safety standards; the sampling adequacy measure was 0.515, indicating that present data were suitable for factor analysis. Bartlett’s test indicated a small p-value less than 0.05 of the significance level, indicating that factor analysis may be useful for this data. The internal consistency of each dimension was checked using Cronbach’s alpha. Cronbach’s alpha factors of 0.60 and above are considered good [17]. 

### 2.4. Validity and Reliability

A literature review on health and safety compliance in the mining industry was conducted to ensure validity. Face validity was ensured through consultation with the health and safety practitioners to confirm that the questionnaire was, indeed, measuring what it was designed to measure [18]. Cronbach’s alpha is one of the most internal consistent measurements that affords a measure of the extent of the items on a measurement scale or internal correlations of items [19]. Reliability was ensured by pilot testing the translated questionnaire, which yielded a Cronbach’s value of 0.930. 

## 3. Results

### Descriptive Statistics

Table 1 shows the general characteristics of the study respondents. The majority were males (89%) aged between 39 and 48. About 54% of the participants were single, while 44% were married. About 42% of the participants indicated that their highest level of education was high school, followed by a college (41%) and primary school (7%). The majority of respondents had between 1 and 10 years of experience. Most of the participants were full-time employees (84%).

The total variance explained by the five factors was 71.4%, suggesting a satisfactory factor solution. The PCA revealed that seven factors could measure occupational health and safety standards practices, as shown in Figure 1. These seven factors were named according to their composition of items: 

Factor 1: Occupational health practice related to regulations; 

Factor 2: Measures to reduce risk of injuries/accidents; 

Factor 3: Role of safety culture and resources; 

Factor 4: Impact of the environment and production; 

Factor 5: Attitude and behaviour; 

Factor 6: The role of mine management; and

Factor 7: Use and quality of equipment (see Table 2).

Table 2 illustrates standardised loading extraction with Varimax rotation conducted for occupational health and safety standards compliance items, to display the convergence of the constructs compromising on a single factor. The PCA commenced with the Kaiser–Meyer–Olkin test for compliance with occupational health and safety standards; the sampling adequacy measure was 0.515, indicating that present data were suitable for factor analysis. Bartlett’s test indicated a small p-value less than 0.05 of the significance level, indicating that factor analysis may be useful for this data. Both tests indicate a significant correlation between the variables and support the relevance of factor analysis [19].

The Cronbach’s alpha for the 29 items was 0.911 (see Table 2). The composite measure of internal consistency was above 0.60 for all factors, suggesting that all constructs were reliable, except Factor 7. Among the elements, knowledge and the availability of health and safety standards, procedures, and policy elements were perceived as fairly high, with a mean of 1.72 and a standard deviation of 0.855; meanwhile, the language used to publish policies, standards, and instructions at work stations were perceived as rather low with a mean score of 1.68 and a standard deviation of 0.841. As indicated by the results, the mean response of employees regarding occupational health and safety practices at the mine was between 1.68 and 1.72, thus indicating that most participants agreed that occupational health is practiced according to the regulations. The Average Variance Extracted (AVE) values were above the recommended benchmark of 0.5, except those for Factor 7. We can conclude that the convergent validity was deemed acceptable given the study’s exploratory nature [19,20] (see Table 3). 

Table 4 illustrates the standardised component correlation matrix for occupational health and safety standards practices in the mining industry. It was observed that Factor 1 (occupational health practice related to regulations) is correlated with Factor 2 (measures to reduce risk of injuries/accidents), and Factor 2 (measures to minimise the risk of injuries/accidents) is correlated with Factor 4 (impact of the environment and production).

## 4. Discussion

This study sought to explore practices of the occupational health and safety legislation/standards compliance in the South African mining industry, Limpopo Province. The study revealed that most respondents were male, at 74%, with only 26% being female. The issue of gender inequality has been under discussion in the SA mining industry, with provisions in place for the inclusion of women in core mining activities. Though some progress has been made in employing more women, with the requirement of 10% of positions to be occupied by women, mining has always been considered a very masculine industry due to its heavily male-dominated workforce and the physicality of mining work [21]. In SA, a study conducted by Zungu [22] revealed that for most women in SA, the mining industry has not been an obvious career choice. This is mostly due to safety concerns in the mines. A total of 92.4% of respondents reported that they were working in the mine due to a lack of other job opportunities. Botha and Cronje [21] argue that the improvement in health and safety standards in SA mining has the potential to provide for the inclusion of women in the mining industry. Hence, the current study aimed to explore the practices related to compliance with health and safety standards. 

The miners who were considered to be more experienced (18%) had more than 10 years of working experience compared to those with 1 to 5 years (36%) experience. Haas, Eiter, Hoebbel, and Ryan [23] conducted a study on the impact of the job, site, and industry experience on worker health and safety. They concluded that miners who are more experienced have a higher level of knowledge in their job, and also, that the level of compliance with health and safety regulations is higher. Klein and Duplessis [2] also supported that miners with less experience are more likely to participate in sub-standard practices because they have not previously experienced dangerous situations or witnessed loss of life as a result of non-compliance. The majority of the respondents were employed on a full-time basis (84%) and 16% of the miners were employed either on a part-time or temporary basis. Only 52% of the miners had tertiary qualifications, either a college certificate or university degree; 36% had a high school education; and 10% had primary school education, while 3% never attended school. A study conducted by Shibambu [8] outlined that because, historically, South Africa is an underprivileged country, most of the people there are not educated, especially the older generation; therefore, they lack skills and qualifications. 

### 4.1. Measurement Model Results for Occupational Health and Safety Standards Compliance

#### 4.1.1. Factor 1: Occupational Health Practice Related to Regulations

Factor 1 included items that addressed the importance of, and practices related to, the occupational health legislation and standards, and their relation to health and safety compliance within the mining industry. By looking at the items within this factor, it is clear that knowledge, availability, implementation, and training on health and safety regulations are viewed as important in ensuring health and safety compliance. The results of this study are similar to previous studies, which revealed that knowledge and the accessibility of occupational health and safety legislation impact implementation and compliance amongst miners [6]. More importantly, adherence to health and safety standards/policies is influenced by the practice of knowledge exchange systems and the degree of knowledge exchange in the organisational system, both within and between units [24]. This is consistent with the study findings, as it was found that training and the availability of policies/standards and procedures have a greater impact on how the miners perceive compliance with the health and safety standards. Hence, miners who have insight into health and safety procedures and policies are more likely to comply with those procedures and change their behaviour to promote safety. 

#### 4.1.2. Factor 2: Measures to Reduce the Risk of Injuries/Accidents

Factor 2 included all items related to the existing measures available to reduce the mine’s risks of injuries and accidents. It is also recognised that the mine has developed specific health and safety standards related to miners’ duties. It was observed that most accidents in the mining industry are caused by non-compliance with health and safety standards. In agreement with the study findings, the existing literature reveals that most accidents and injuries are caused by non-compliance with the available regulations and standards and the organisation’s established health and safety management system [25]. Therefore, the study findings suggest a need for the mines to develop measures to create a high standard of safety compliance, to reduce the risk of injuries/accidents. The existing health and safety standards cover all the perceived risks. This mine item recorded the highest loading under this dimension, suggesting that this item contributes the most to reducing the risk of injuries/accidents. Improving existing safety standards in the mines remains a priority to enhance practice and eliminate risk. 

#### 4.1.3. Factor 3: Role of a Safety Culture and Resources

Factor 3 included all items describing the role of a safety culture and resources in ensuring safety compliance. According to Widajati, Ernawati, and Martiana (2017) [26], the safety team has an influential role in ensuring a work environment that is free from accidents and injuries. It was revealed from the current study findings that the health and safety team (health and safety officers, representatives, and occupational health practitioners) play an essential role in ensuring health and safety standards compliance in this mine. The occupational health clinic and the safety department were viewed as the core departments in facilitating and supporting safety activities in the mine. This was followed by the rating of the union and other different organisations in the mine, which were noted to influence the behaviour of miners towards compliance with the safety standards. It was also noted that the culture created in this mine is one that values the organisation’s profit more than the safety of the miners. All the items generated gave more agreeable than disagreeable outcomes. Health and safety teams play an important role in ensuring safety during mine accidents. The research item recorded the highest loading under this dimension, suggesting that this item contributes the most to the role of safety culture and resources.

#### 4.1.4. Factor 4: Impact of the Environment and Production

Factor 4 highlights elements describing the impact of the environment and production on compliance with the health and safety standards. The physical environment is a poorly designed item that recorded the highest loading under this dimension, suggesting that this item contributes the most to the impact of the environment and production. Non-compliance with safety standards could, thus, result from a poorly designed physical environment. Qasim et al. [27] conducted similar scholarly work to the current study findings, and emphasised that safety is associated with the physical mining environment and interventions performed to reduce risk exposure. Furthermore, Cooper [28] reported that safe production implies that every person is aware of and values that the organisation’s prosperity, and survival depends on safe production. The current study findings suggest that adopting a safe production strategy, such as effectively designing a physical mining environment, can set the health and safety bar high. The Cultural Transformation Model [29] emphasises the importance of the mining organisation and its structure, always remembering that safety and production are not competing for objectives, but rather, an outcome of the work well carried out.

The literature revealed production pressure as the miners’ fundamental cause of poor compliance, especially for those who intend to meet the production targets, which results in “doing things faster” while compromising their health and safety [30]. This literature shares the same sentiments with the current study findings, which reported production pressure as a second priority item that results in poor compliance and a high risk of injuries. 

#### 4.1.5. Factor 5: Attitude and Behaviour

Factor 5 included the attitude and behaviour items associated with safety compliance. According to Li et al. [31], a safety attitude can positively influence safety behaviour, thus suggesting that adopting a safety culture can improve safety compliance. A study conducted by Haas, Eiter, Hoebbel, and Ryan [23] concluded that miners who are more experienced have a higher level of knowledge of their job and that their level of compliance with health and safety is proportionately higher. This is in agreement with the current study findings, which indicate that young employees are more likely to engage in sub-standard practices. Klein and Duplessis [2] (2016) also supported that miners with less experience are likely to participate in sub-standard practices because they have not previously experienced dangerous situations or witnessed loss of life due to non-compliance. A study conducted by Nordlof et al. [32] (2015) indicated that less-experienced employees who are new to the workplace are more likely to take risks. Therefore, the level of experience in the mining industry negatively impacts safety compliance. Most of the participants were below the age of 50, and only 19% were above the age of 50. The varying ideas, values, and experiences of a different generation can affect health and safety compliance in the workplace [33]. The highest-loading item about attitude and behaviour relates to employees’ age. For example, the current study findings indicate that young employees are more likely to engage in sub-standard practices. According to the South African Small-Scale Mining guidelines [34], mining organisations must strengthen positive behaviour without considering the employees’ age to promote a crucial health and safety culture for risk control in the workplace. Hence, it is important to note that safety compliance is not mere compliance with regulations and standards, but a culture that compels the miners and organisation to take responsibility for their behaviour and actions towards health and safety.

#### 4.1.6. Factor 6: The Role of Mine Management

Factor 6 included all items indicating the mine management’s significance in influencing miners’ health and safety compliance. The aspect that stands out is that violation/non-compliance of health and safety regulations causes accidents. This item recorded the highest loading under this dimension. This suggests that mine management plays a significant role in promoting compliance, including monitoring and supervising miners during production activities. The current study’s findings are echoed in scholarly work wherein the role of leaders in supervision affects the safety practices in the workplace and provides an opportunity to influence positive outcomes for workers, managers, and the organisation [35]. Practising supervision by mine management could mean that management makes honest and reasonable efforts to promote a healthy working environment [4], which is loaded as a second priority for the role of mine management in the current study findings.

#### 4.1.7. Factor 7: Use and Quality of Equipment

Factor 7 included all items on the use and quality of the equipment that influences health and safety compliance. The item with the highest loading was “equipment provided is good quality”. The current study findings are in contrast to that of Shaw, Verna Blewett, and Schutte [35], who outlined that most mining industries failed to invest in technology, reducing the practicability of risk. In addition to the quality equipment provided, there is an acknowledgement relating to the procedures available for regular inspection of such equipment. The current study findings commend the role of equipment and its management in the mining industry.

### 4.2. Correlated Components

#### 4.2.1. Factor 1 (Occupational Health Practice Related to Regulations) Is Correlated with Factor 2 (Measures to Reduce Risk of Injuries/Accidents)

The Pearson correlation test suggests a positive correlation between the occupational health practice regulations and measures to reduce the risk of injuries/accidents. Our study findings are related to those reported by the WHO Healthy Framework, which states that each mining organisation must strive to achieve compliance and create a healthy work environment. This could be achieved by involving the miners in focus groups so that they can identify and implement the solutions in unity [36]. Our findings suggest that reducing risks yields good practice among miners. For example, creating support groups among the miners to minimise the risk of injuries may increase adherence to health and safety instructions. There is a correlation between compliance with the safety regulations and the measures to reduce injuries and accidents.

#### 4.2.2. Factor 2 (Measures to Reduce the Risk of Injuries/Accidents) Is Correlated with Factor 4 (Impact of the Environment and Production)

Measures to reduce the risk of injuries/accidents, and the impact of the environment and production on safety compliance, had the strongest correlation. This implies that the impact of an environment, such as the elimination of the risk factors for injuries within the mining environment, could result in a better presentation of the mining industry. The current study findings are similar to those reported in the existing literature. For example, Michell [37] reported that by ensuring that compliance with health and safety requirements is adhered to, occupational health practitioners must increase awareness of the health risks that impact people’s health and safety and the measures that can mitigate the dangers. Cooper [28] argued that safe production implies that every person is aware of and values the fact that the organisation’s prosperity and survival depend on safe production, and where there is conflict, safety becomes a priority. The study results indicate that the mining environment’s activities and production pressure may affect and compromise miners’ safety and, subsequently, non-compliance with safety guidelines.

## 5. Limitations of the Study

The COVID-19 pandemic affected the data collection process, making it difficult for the researchers to access the respondents, which affected the study’s sample size.

## 6. Conclusions and Recommendations

This study contributes to the body of knowledge about the practices of occupational health and safety legislation/standards compliance. The study results indicate that the mining environment’s activities and production pressure may affect and compromise miners’ safety and, subsequently, result in non-compliance to safety guidelines. Therefore, a safety awareness/behaviour awareness programme is recommended to encourage the miners’ behaviour in promoting safety in the working environment and during production, which will be healthy and safer for everyone. The study also revealed that training and the availability of policies/standards and procedures has a greater impact on how the miners perceive compliance with the health and safety standards. Strengthening the training and education system by adopting new education methods using information and communication technologies, sharing information through social networks, using websites, and e-learning will be beneficial in creating a platform for miners to quickly discuss the issue of health and safety at work. It is also recommended that the mining organisations must develop and implement an effective health and safety management system that contributes to building a positive health and safety culture at work. Moreover, understanding the health and safety practices of miners might assist the mine in developing on-site interventions to improve the state of health and safety.

## Figures and Tables

**Figure 1 ijerph-19-10241-f001:**
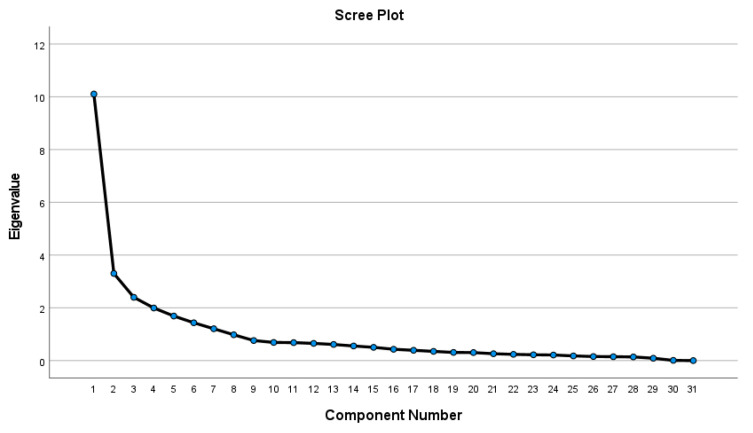
Factors Scree Plot for occupational health and safety standards practices in the mining industry.

**Table 1 ijerph-19-10241-t001:** Demographic details of participants.

Variable	Characteristics	Frequency	Percentage
Sex	Male	246	89
	Female	31	11
Age	20–28	70	25
	29–38	74	27
	39–48	79	29
	49–58	42	15
	59–65	12	4
Marital Status	Single	152	54
	Married	120	44
	Divorced	5	2
Ethnicity	Black	270	97
	White/coloured	7	3
Highest level of education	College	114	41
	Technikon	14	4
	University	13	5
	High school	116	42
	Primary	20	7
Years of experience	Under 1 year	32	12
	1–5 years	77	28
	6–10 years	80	29
	11–15 years	42	15
	16–20 years	16	6
	Over 20 years	30	11
Employment status	Full-time	231	84
	Part-time	21	8
	Temporary	25	9

**Table 2 ijerph-19-10241-t002:** Measurement model results for occupational health and safety standards compliance in a selected mine, Limpopo Province.

Practice	Loading	Mean ± SD
Factor 1: Occupational health practice related to regulations	α= 0.934	1.67 ± 0.032
Safety regulations and standards play an important role in avoiding accidents at the mines	0.954	1.72 ± 0.855
The occupational health clinic plays a role in promoting the health and safety of employees	0.945	1.72 ± 0.861
I receive full training on health and safety regulations in this mine	0.933	1.66 ± 0.753
I know the regulations designed to protect the health and safety of employees in this mine	0.900	1.67 ± 0.770
Policies for improving health and safety are published at the mine administration	0.673	1.68 ± 0.826
I understand the language used to publish policies, standards, and instructions at my workstation	0.619	1.68 ± 0.841
Factor 2: Measures to reduce the risk of injuries/accidents	α = 0.862	1.71 ± 0.10
The existing health and safety standards cover all the risks in this mine	0.956	1.66 ± 0.778
My employer has developed specific health and safety standards that relate to the work that I do daily	0.843	1.68 ± 0.807
Accidents are caused by non-compliance with the health and safety standards	0.706	1.74 ± 0.809
There are adequate policies for investigating and preventing further accidents	0.695	1.62 ± 0.891
There is a need for the mine to develop measures to create a high standard of safety culture	0.670	1.90 ± 0.661
Health and safety are taken seriously and respected in this mine	0.617	1.68 ± 0.888
Factor 3: Role of safety culture and resources	α = 0.827	1.98 ± 0.09
Health and safety teams play an important role in ensuring safety in mine accidents and the mines in general	0.997	1.94 ± 0.832
The unions and different organisations in the mine influence the behaviour of employees towards compliance with the standards	0.997	1.94 ± 0.832
The culture created in this mine is to value the profits of the company above the safety of the miners	0.679	2.15 ± 0.961
Lack of resources and proper equipment causes non-compliance	0.620	1.98 ± 0.829
People-oriented safety culture is practiced and considered at this mine	0.494	1.90 ± 0.854
Factor 4: Impact of the Environment and production	α = 0.779	1.98 ± 0.013
The physical environment is poorly designed	0.932	2.12 ± 0.838
The production pressure contributes to the non-compliance with health and safety standards	0.770	2.03 ± 0.863
Accidents are caused by a temporary unsafe environment created as a result of the work process	0.723	1.81 ± 0.829
The environment is safe and free from risks such as heat, noise, slippery floors, and poor ventilation	0.702	1.96 ± 0.733
Factor 5: Attitude and behavior	α = 0.796	2.10 ± 0.002
The young employees are more likely to engage in sub-standard practices	0.878	2.09 ± 0.863
There is an association of employees’ cultural beliefs or religious background with adherence to the health and safety standards	0.787	2.10 ± 0.830
The attitude and behaviour associated with unsafe acts or compliance is related to the mineworker’s experience	0.732	2.11 ± 0.796
Factor 6: The role of mine management		2.146 ± 0.386
Violation/Non-compliance with health and safety regulations causes accidents	0.764	2.42 ± 1.125
The mine management makes honest and reasonable efforts to promote a healthy working environment	−0.742	1.87 ± 0.610
Factor 7: Use and quality of equipment	α = 0.227	1.81 ± 0.161
The equipment provided is of a good quality	0.620	1.82 ± 0.763
There are procedures available for the regular inspection of equipment	0.618	1.97 ± 0.782
Tokens of appreciation for lack of injuries/accidents can improve compliance	−0.502	1.65 ± 0.752
Overall Alpha	α = 0.911	
Kaiser–Meyer–Olkin Measure	0.615	
Bartlett’s Test	682.67	*p*-value < 0.001

**Table 3 ijerph-19-10241-t003:** Average Variance Extracted (AVE) measurement.

Construct	Average Variance Extracted
Factor 1: Occupational health practice related to regulations	0.572
Factor 2: Measures to reduce the risk of injuries/accidents	0.572
Factor 3: The role of safety culture and resources	0.615
Factor 4: The impact of the environment and production	0.619
Factor 5: Attitude and behavior	0.642
Factor 7: Use and quality of equipment	0.339

**Table 4 ijerph-19-10241-t004:** Component correlation matrix for occupational health and safety standards practices in the mining industry.

Component	Factor 1	Factor 2	Factor 3	Factor 4	Factor 5	Factor 6	Factor 7
Factor 1	1.000	0.566	0.354	0.410	0.160	0.244	0.090
Factor 2	0.566	1.000	0.410	0.526	0.074	0.251	0.108
Factor 3	0.354	0.410	1.000	0.405	0.182	0.120	0.194
Factor 4	0.410	0.526	0.405	1.000	0.184	0.055	0.125
Factor 5	0.160	0.074	0.182	0.184	1.000	−0.262	−0.069
Factor 6	0.244	0.251	0.120	0.055	−0.262	1.000	0.028
Factor 7	0.090	0.108	0.194	0.125	−0.069	0.028	1.000

## Data Availability

Data generated and analyzed during the current study are no publicly available due to ethical reasons but are available from the corresponding authors.

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
