# Peer review of "Exploring Occupational Health and Safety Standards Compliance in the South African Mining Industry, Limpopo Province, Using Principal Component Analysis"

_ijerph, 2022, doi:10.3390/ijerph191610241_

Round 1
Reviewer 1 Report
In the manuscript ijerph-1679810 authors presented the results of a study aimed to explore the practices related to occupational Health and Safety standards Compliance in the South African Mining Industry. A quantitative descriptive design was applied. Data was collected using a survey questionnaire administered to miners to explore the Occupational Health and Safety standards Compliance. Principal Components Analysis (PCA) extracted key attributes to Occupational Health and Safety standards Compliance in the mining industry and to assess relationships between different dimensions.
Overall, the manuscript should be improved in terms of clarity of exposure and exhaustiveness. Some methodological details should be provided, which I ask the authors to clarify.
- Please consider conducting a careful spell-check and text review to eliminate typos and improve text formatting.
- Section 2.2. Population and Data collection: Please consider providing a copy (in English) of the questionnaire and / or reporting an ordered list of requested variables and possible responses.
- 2.3. Data analysis. Please provide further information. number of variables specific PCA procedures, etc.
- Section 2.4. Validity and reliability.
More than a support from the bibliography, I would expect here some performance analysis of the PCA model and / or a sensitivity analysis and / or uncertainty assessment procedure - Please check information in table 1: specify what is "Technikon";
- The approach used to present the results however is quite convincing and quite effective. I only suggest working on clarity of exposure and optimization of contents
Author Response
TABLE OF CORRECTIONS
REVIWER 1
|
Comments
|
Remarks |
Page No |
|
1. Overall, the manuscript should be improved in terms of clarity of exposure and exhaustiveness. Some methodological details should be provided, which I ask the authors to clarify. |
Clarified |
Methodology |
|
2. Please consider conducting a careful spell-check and text review to eliminate typos and improve text formatting. |
The article was sent for editing |
All pages |
|
3. Population and Data collection: Please consider providing a copy (in English) of the questionnaire and / or reporting an ordered list of requested variables and possible responses. |
Copy of questionnaire attached |
Appendix 1 |
|
4. Data analysis. Please provide further information. number of variables specific PCA procedures, etc. |
Corrected |
2.3 Line 3 to 4 |
|
Corrected |
2.3. line 8 - 12 |
|
|
6. Please check information in table 1: specify what is "Technicon"; |
Corrected table 1 |
Page 3 |
|
Corrected |
6-10 |

Reviewer 2 Report
The authors conducted a study focusing on the compliance status of miners in the local mining industry. My top concern is the writing skills (lots of details can be found below). Many sentences become less meaningful if the words were not carefully selected. My second concern is the way the author interprets the results. The results were based on subjective feedback from the participants instead of actual numbers. The discussion and conclusion should focus on the results and why instead of claiming they are the truth. E.g. No data supports non-compliance is highly correlated with occurrences of injuries and accidents. It’s the participants thinking that they are highly correlated.
Introduction
First paragraph, line 4: Extra comma before the citation
Second paragraph, line 4: Semicolon should be colon
Second paragraph, line 9: Not sure why quotation is used for “improvement of the safety standards in the mining sector”.
Second paragraph, line 10: The word “Thus” indicates a direct logical correlation between “legislation approved South African mining industry assists the transformation” and “miners’ health and safety standards are rights”. However, I don’t see obviously logical relations.
The last sentence of the second paragraph confuses me, and it’s hard to interpret what the author wants to address.
Third paragraph, line 2: “SA” should be introduced when the full words first appear in the content, e.g.: “South African (SA)”
Third paragraph, line 3: The word “However” doesn’t seem to fit here. The previous and afterward sentence doesn’t contradict each other but supports each other on some level.
Methodology
Section 2.1 First paragraph, line 1: “design was used during obtaining data” is probably better
Section 2.2 First paragraph, line 1: The word “willingness” is very vague. Sampling based on “willingness and availability” could create bias in the statistical results and cannot be called “randomly selected” in the abstract. E.g. People who show more compliance to the policies and rules may be more willing to participate in this type of study, answer and return the questionnaire.
Section 2.2 First paragraph, line 11: Three sections were stated though only Sections A&B were introduced
Section 2.3 First paragraph, line 7: “EFA” is not described previously
Section 2.3 First paragraph, line 9: “these factors and above are considered good” is better
Section 2.4 Frist paragraph, line 2: The sentence “To ensure face validity …” is not understandable. Consider rephrase
Results
Figure 1 was referred to after Table 2 but presented before Table 2
Table and figure qualities need improvement
Discussion
While stating 74% male and only 26% female responded, the authors should carefully think if this is really related to the inclusion of women in core mining activities. There are lots of possible arguments around this. E.g. more males and fewer females in the industry may be the result of females’ unwillingness of entering the industry instead of inclusion issues. If inclusion is actually the case, the authors should provide more evidence/citations
Second paragraph line 1: First sentence got grammar mistakes
Second paragraph line 2: The citation number should be next to the last author’s name of the publication
Instead of stating the gender/experience/time-basis percentage, I’m more concerned about how well the sample size composition represents the actual miner population. For example, was the male-female ratio in the industry similar to the ratio in the study sample?
I think the abstract and conclusion are misleading. In the abstract, the author said, “There is a high correlation between non-compliance with the safety regulations and the occurrence of injuries and accidents”. This is a very important and significant finding. However, after reading the content, the actual correlation was between the study participants’ perceptions of the consequences of non-compliances. The results are very subjective and might just be a result of policy/rule awareness. The results can only suggest that it’s common sense among the participants that non-compliance with the safety regulations is correlated with the occurrence of injuries and accidents. This exactly contradicts the author's statement in the abstract conclusion: ”A safety behavior awareness program is recommended to help …”
Author Response
REVIWER 2
|
Comments
|
Remarks |
Page No |
|
1. My top concern is the writing skills (lots of details can be found below). Many sentences become less meaningful if the words were not carefully selected. |
The article was sent for editing |
All |
|
2. My second concern is the way the author interprets the results. The results were based on subjective feedback from the participants instead of actual numbers. |
In the discussion of the results, the authors tried to interpret the results as actual number |
Page 6-9 |
|
3. The discussion and conclusion should focus on the results and why instead of claiming they are the truth. E.g. No data supports non-compliance is highly correlated with occurrences of injuries and accidents. It’s the participants thinking that they are highly correlated. |
Corrected |
Page 6-10 |
|
4. Introduction · First paragraph, line 4: Extra comma before the citation · Second paragraph, line 4: Semicolon should be colon · Second paragraph, line 9: Not sure why quotation is used for “improvement of the safety standards in the mining sector”. · Second paragraph, line 10: The word “Thus” indicates a direct logical correlation between “legislation approved South African mining industry assists the transformation” and “miners’ health and safety standards are rights”. However, I don’t see obviously logical relations. · The last sentence of the second paragraph confuses me, and it’s hard to interpret what the author wants to address. · Third paragraph, line 2: “SA” should be introduced when the full words first appear in the content, e.g.: “South African (SA)” · Third paragraph, line 3: The word “However” doesn’t seem to fit here. The previous and afterward sentence doesn’t contradict each other but supports each other on some level.
|
All suggestions effected |
Page 1-2 |
|
All suggestions effected |
Page 2-3 |
|
|
6. Results · Figure 1 was referred to after Table 2 but presented before Table 2 · Table and figure qualities need improvement
|
Corrected |
Page 4 |
|
More citations provided. The study focus was not on comparing he gender variable to the experience. Hence the current study aimed to explore the practices related to compliance with the health and safety standards. |
Line 9 to 15 |
|
|
Abstract was aligned with the results and conclusion |
Page 1 |

Round 2
Reviewer 1 Report
The authors have responded adequately to all comments and the revised version has greatly improved. I have no other comments to submit.
Reviewer 2 Report
I'm glad to see the significant improvements in the manuscripts made by the authors. I only have suggestions on minor things (e.g. formatting):
The font style in Figure 1 should match the font style in the content.
The discussion about each factor is important. It would be better if each factor discussion becomes a subsegment with subtitles using Italic font (similar to 2.1)